# The Effect of Accommodation on Peripheral Refraction under Two Illumination Conditions

Raquel van Ginkel [1,*], María Mechó [2] , Genis Cardona [3] and José M. González-Méijome [2]

1 School of Optics and Optometry of Terrassa, Universitat Politècnica de Catalunya, BarcelonaTech, Violinista Vellsolà 37, 08222 Terrassa, Spain
2 Clinical and Experimental Optometry Research Lab (CEORLab), Center of Physics, School of Sciences (Optometry), University of Minho, 4710-057 Braga, Portugal; mmechogarcia@fisica.uminho.pt (M.M.); jgmeijome@fisica.uminho.pt (J.M.G.-M.)
3 Optics and Optometry Department, Universitat Politècnica de Catalunya, BarcelonaTech, Violinista Vellsolà 37, 08222 Terrassa, Spain; genis.cardona@upc.edu
* Correspondence: raquel.van.ginkel@estudiantat.upc.edu

**Abstract:** The clinical importance of peripheral refraction as a function of accommodation has become increasingly evident in the last years with special attention given to myopia control. Low order ocular aberrations were measured with a Hartmann–Shack aberrometer in a sample of 28 young emmetropic subjects. A stationary Maltese cross was presented at 2.5 D and 5.0 D of accommodative demand and at $0°$, $10°$ and $20°$ of eccentricity in the horizontal visual field under two different illumination conditions (white and red light). Wavefront data for a 3 mm pupil diameter were analyzed in terms of the vector components of refraction (M, J0 and J45) and the relative peripheral refractive error (RPRE) was calculated. M was myopic at both accommodative demands and showed a statistically significant myopic increase with red illumination. No significant change in J0 and J45 was found with accommodation nor between illumination conditions. However, J0 increased significantly with eccentricity, exhibiting a nasal-temporal asymmetry. The RPRE was myopic at both accommodation demands and showed a statistically significant hyperopic shift at $20°$ in the nasal retina. The use of red light introduced statistically and clinically significant changes in M, explained by the variation of the ocular focal length under a higher wavelength illumination, increasing the experimental accommodative demand. These findings may be of relevance for research exploring peripheral refraction under accommodation, as the choice of target illumination is not trivial.

**Keywords:** peripheral refraction; accommodation; aberrometry; relative peripheral refractive error; IRX3 aberrometer





## 1. Introduction

Myopia control is one of the main challenges in vision science and visual health for the 21st century. The prevalence of myopia has risen over the past 60 years [1] and by 2050 it is estimated that 50 percent of the global population will be myopic [2].

The mechanisms that govern the development and evolution of myopia after birth are not fully understood, but a number of factors have been implicated in this development, including genetic predisposition [3,4] environmental conditions [5], time spent outdoors or conducting near work tasks [6,7]. Indeed, previous animal studies have shown that the axial growth of the eye and its refractive status after birth can be influenced by altered visual demand [8].

Research efforts have been directed to the relation between non-foveal (peripheral or out of axis) and central refractive error, evidencing a hyperopic shift in the peripheral retina in myopic eyes, related to an increase in axial length [8–13]. These studies concluded that peripheral hyperopic defocus (behind the retina related to the fovea) may be a stimulus for eye growth, leading to axial myopia.

A range of myopia management and control strategies are currently available, including orthokeratology, multifocal contact lenses and multifocal or progressive ophthalmic lenses [14,15]. Whereas conventional ophthalmic and contact lenses correct foveal refractive error, the optical methods and strategies designed for myopia progression control aim at providing an additional modification in the peripheral image quality or focus [16–18]. However, the actual mechanisms leading to myopia onset and progression, and underpinning the performance of these strategies, remain controversial.

Changes in peripheral aberrations as a function of accommodation have received little attention. Previous studies have evaluated on-axis optical aberrations as a function of accommodation [19] or have assessed optical peripheral aberrations in relaxed eyes under natural or cycloplegic conditions [20–22]. Only a few studies have assessed peripheral aberrations with accommodation, with inconclusive findings [23–29]. Furthermore, accommodative range, eye gaze position, instrumentation, pupil diameter and visual stimulus vary among the studies, making a comparison of results difficult.

The purpose of the present research was to develop an experimental design to measure on-axis and off-axis low order optical aberrations at different levels of accommodation with a commercial Hartmann–Shack aberrometer and to compare the results obtained under two illumination (red and white light) conditions in a sample of healthy, young and emmetropic subjects. Data for a 3 mm diameter pupil was analyzed in terms of the vector components of refraction (M, J0 and J45) and the relative peripheral refractive error (RPRE) was calculated for each eccentricity. Stimuli were presented at 2.5 D and 5 D of accommodative demand and at 0°, 10° and 20° of eccentricity in the horizontal visual field (nasal and temporal).

## 2. Materials and Methods

### 2.1. Subjects

Subjects aged between 18 and 35 years were recruited from the Universidade do Minho during the months of November and December 2021. All subjects were free of ocular pathologies and had uncorrected visual acuity (VA) of 0.0 logMAR or better. The presence of accommodation disorders, previous refractive surgery or other ocular interventions, systemic medication, contact lens wear, or dry eye symptomatology were grounds for exclusion.

Informed consent was obtained from all participants, following an explanation of the study procedures in accordance with the tenets of the Declaration of Helsinki. Ethical approval was obtained from the Scientific Committee of the School of Science of Minho University (Portugal).

### 2.2. Materials

The IRX3 commercial Hartmann–Shack aberrometer (Imagine Eyes, Orsay, France) was used in this study. This instrument has a $32 \times 32$ lenslet array and uses 780 nm wavelength for aberrometry measurements. The IRX3 aberrometer has been used in previous studies [30–33], one of which aimed at assessing peripheral aberrations at 20° of eccentricity in the horizontal visual field [33]. The refractive error range that can be measured with this instrument is $-15$ D to $+20$ D, by means of the sphere and, $-10$ D to $+10$ D, by means of the cylinder.

The internal viewing target of the IRX3 aberrometer is designed for central measurements and consists of a black 6/12 Snellen letter "E" over a white elliptical background field subtending about $0.7 \times 10$ degrees, with a luminance of 85 cd/m$^2$. In order to obtain peripheral measurements along the horizontal visual field, a modified target system was used with an additional beam splitter inserted between the eye and the aberrometer. The beam splitter allowed the projection of peripheral and central targets placed in a superior position to the tested eyes. For this purpose, only the right eye (RE) was used in this study.

Two methacrylate accessory plates with two symmetrical front slots at two different distances were attached to the aberrometer. The symmetrical front slots allowed to place other accessory methacrylate plates to present the targets (Figure 1).

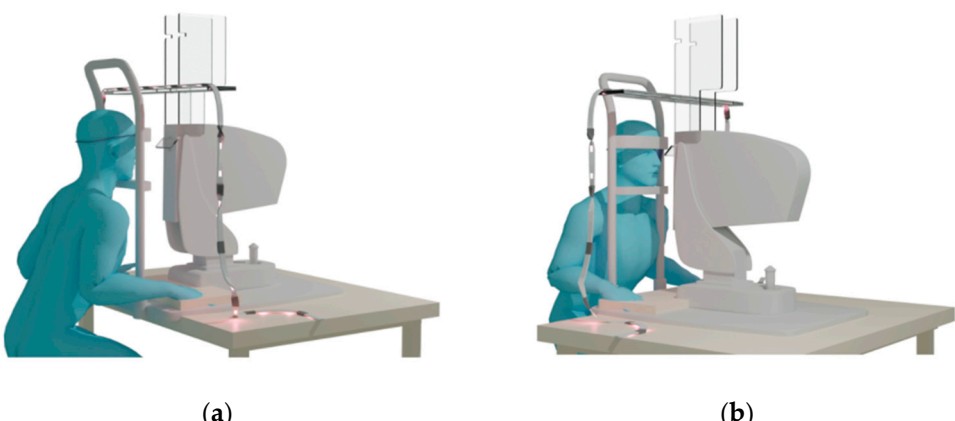

(**a**)          (**b**)

**Figure 1.** A 3D representation of the experimental design: (**a**) Lateral-posterior view; (**b**) Lateral-frontal view.

The experimental setup was designed in order to measure optical aberrations in the horizontal visual field at two accommodative demands: 5 D (20 cm) and 2.5 D (40 cm). For each accommodative distance, a horizontal band with five accommodative stimuli was designed. A central stimulus was placed at 0° and two peripheral stimuli were placed at 10° and 20° on each side of the central stimulus. The accommodative stimuli consisted of a black Maltese cross subtending 0.43° in diameter at both viewing distances and were printed on translucent vellum paper in order to avoid light reflections and to allow light to pass through. For each accommodative distance, an identical second set of stimuli was placed in the methacrylate plates and was backlighted with a red LED light (Figure 2).

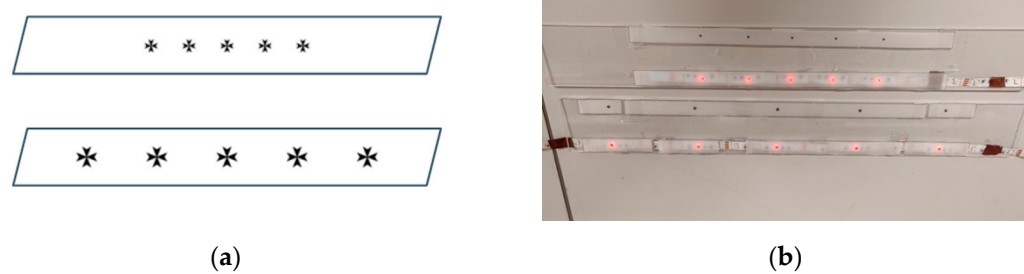

(**a**)          (**b**)

**Figure 2.** Methacrylate accessory plates: (**a**) Superior plate presenting the stimuli for 5 D of accommodative demand and inferior plate presenting the stimuli for 2.5 D of accommodative demand; (**b**) Final accessory plates with an identical second set of red LED backlighted stimuli for each accommodative distance.

The red LED light used in this setup was characterized in terms of the relative emission spectrum, with a wavelength range between 590 nm and 655 nm and the emission peak at 630 nm. When the subjects observed the red LED backlighted stimuli, the laboratory illumination was turned off and the luminance of the stimuli as viewed through the beam splitter was 1.6 cd/m$^2$. In contrast, when the aberrometric measurements were conducted with the laboratory illumination turned on, the luminance of the stimuli as viewed through the beam splitter was 15.2 cd/m$^2$. The purpose of evaluating different combinations of target illumination and background conditions was to ease the fixation task by the subjects and maximize the pupil size (minimize the photomotor reflex) to obtain aberrometric information over a larger pupil area.

### 2.3. Methods

All experimental measurements were conducted in one session and were divided into two parts.

The first part consisted of a full optometric examination, including anamnesis, monocular uncorrected VA at 4.0 m (ETDRS chart, Precision Vision), autorefractometry, subjective refraction and the measurement of Donders amplitude of accommodation. A Maltese cross, identical to the smallest stimuli in the experimental design, was used for the measurement of Donders amplitude of accommodation and the mean of three repeated measures was used for the analysis.

For the second part, ocular aberrations and pupil diameters were assessed in the RE with subjects seated and using a chin and a headrest. Subjects were instructed to remain stationary with both eyes open during the measurements (although an occluder was placed in front of the left eye) and to fixate on the internal target of the aberrometer. While maintaining normal blinking, the pupil was aligned with the instrument axis and baseline measurements (without accommodation) were taken. Then, the accommodative demands of 2.5 D and 5 D were assessed. The sequence of gaze positions started at 0° and was followed by 10° and 20°, beginning with the nasal retina and ending with the temporal retina (Figure 3). Three readings were obtained for each gaze position and, immediately before each measurement, subjects were instructed to blink and then to hold their eyes open. Measurements were first obtained with the laboratory illumination turned on and then were repeated without laboratory illumination and using the red LED backlighted stimuli.

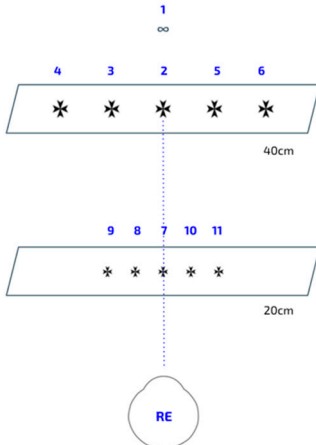

**Figure 3.** Order of the experimental measurements from 1 to 11 (first and last, respectively).

Before each accommodative distance was assessed, subjects were asked to align the central stimulus of the custom-made band with the internal target of the aberrometer. Real captures of the aberrometer alignment screen are shown in Figure 4. Subjects were asked to maintain their head stationary while rotating the eye towards each gaze position.

As the pupil shape becomes elliptical off-axis (Figure 4), a common round pupil size was selected based on the minimal round pupil found in more eccentric gaze positions and for the two target illumination conditions (red, white). Lower order ocular aberrations were extracted for a 3.0 mm pupil diameter for all subjects in all positions. Refraction vector components (M, J0 and J45) were obtained according to Fourier analysis, as recommended by Thibos et al. [34], where Sph, Cyl and θ are the manifest sphere, cylinder and axis, respectively.

$$M = Sph + Cyl/2; \quad J0 = -Cyl \cdot \cos(2\theta)/2; \quad J45 = -Cyl \cdot \sin(2\theta)/2 \tag{1}$$

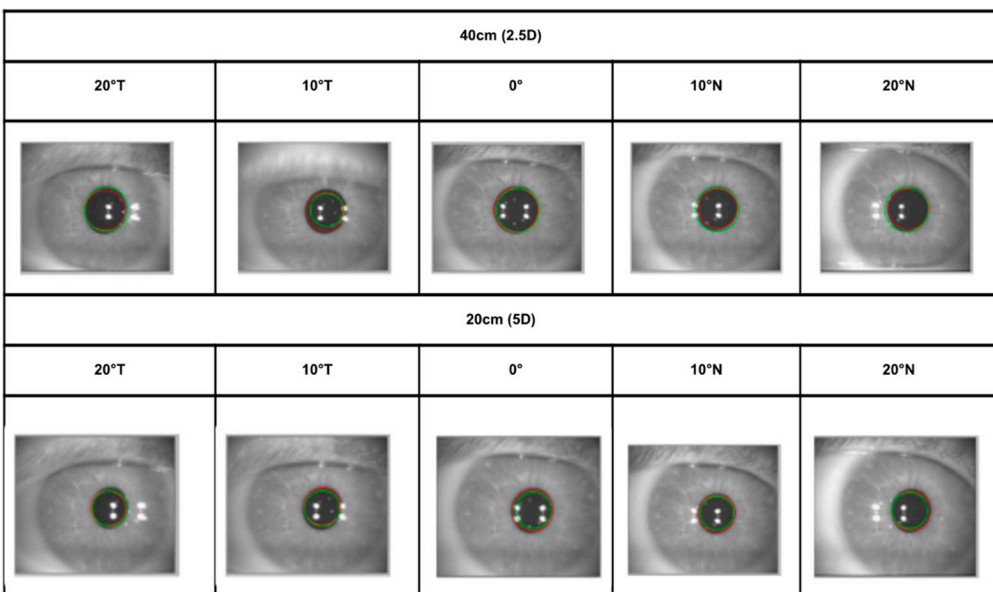

**Figure 4.** Aberrometer alignment screen captures at each gaze position for 2.5 D (**top**) and 5 D (**bottom**) of accommodative demand.

Average refraction vector components of three measurements at each gaze position were calculated and the baseline value (0°, 0 D) was subtracted.

The relative peripheral refractive error (RPRE) for each subject, gaze position and accommodative demand was defined as the averaged spherical equivalent at the central gaze position (0°) subtracted from the averaged spherical equivalent at the corresponding eccentricity.

### 2.4. Statistical Analysis

Statistical analysis was performed using freely available JASP software version 13.0 (University of Amsterdam). The Shapiro–Wilk test was employed to examine data distribution. Descriptive statistics (mean and standard deviation; mean $\pm$ SD) were used to characterize the sample. The Student's *t*-test or the Wilcoxon test were used for paired comparisons of normal and non-normally-distributed data, respectively, and the ANOVA or Friedman tests were used for multiple comparisons, considering the Bonferroni or Conover's correction for post hoc pair-wise analysis. A *p*-value $\leq 0.05$ was defined to denote statistical significance.

## 3. Results

### 3.1. Sample Demographics

Twenty-eight emmetrope students (14 men and 14 women) were enrolled in the present study, with an age range between 19 and 32 years (mean $\pm$ SD; 22.8 $\pm$ 2.8 years). As noted above, the right eye was selected for the purposes of this study, with a mean distance uncorrected visual acuity of 0.11 $\pm$ 0.07 logMAR, mean spherical equivalent of $-0.19 \pm 0.18$ D and mean Donders amplitude of accommodation of 10.82 $\pm$ 1.60 D. The minimum value of the amplitude of accommodation of the study sample was 8.10 D, ensuring the correct visualization of the smallest target presented in the experimental design (5 D).

### 3.2. Spherical Equivalent (M)

The mean spherical equivalent (M) relative to the baseline measurement as a function of eccentricity is shown in Figure 5. Negative values of M in all gaze positions were expected by the relative negative condition (myopic) generated in the accommodated eye as measured with the aberrometer.

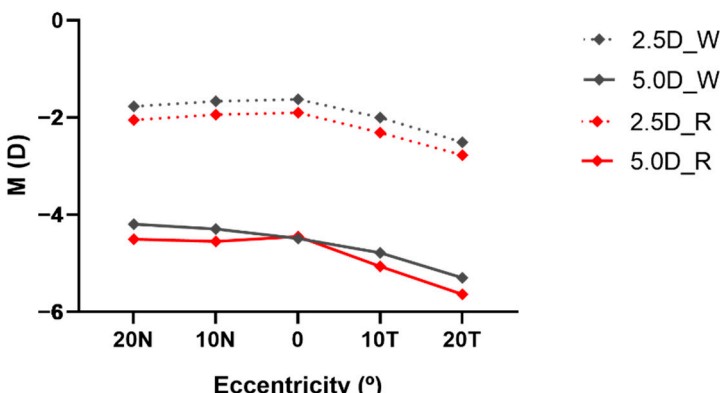

**Figure 5.** Spherical equivalent as a function of eccentricity for 2.5 D (dotted lines) and 5 D (solid lines) accommodative demands (W: white light; R: red light; N: nasal retina; T: temporal retina).

Table 1 summarizes mean spherical equivalent results for each examination condition. The differences obtained between white and red light conditions manifest a statistically significant more myopic M in all gaze positions, except at 0° and 5 D of accommodative demand. The mean difference of M between white and red light, respectively, is $-0.28 \pm 0.02$ D for 2.5 D of accommodative demand and $-0.23 \pm 0.15$ D for 5 D of accommodative demand. Assuming a minimum value of 0.25 D for clinical significance, results can be considered clinically significant in almost all gaze positions.

**Table 1.** Spherical equivalent (mean ± standard deviation) with both illumination conditions (red and white light) at each eccentricity and accommodative demand. (N: nasal retina; T: temporal retina).

| Examination Conditions | | Spherical Equivalent (Mean ± SD) | | | |
| --- | --- | --- | --- | --- | --- |
| | | White Light | Red Light | Difference $_{(W-R)}$ | *p*-Value [§] |
| 2.5 D | Axial | $-1.63 \pm 0.53$ | $-1.90 \pm 0.56$ | $-0.27$ | 0.019 * |
| | 10°N | $-1.67 \pm 0.56$ | $-1.94 \pm 0.70$ | $-0.27$ | 0.002 * |
| | 20°N | $-1.77 \pm 0.79$ | $-2.05 \pm 0.98$ | $-0.28$ | 0.006 * |
| | 10°T | $-2.00 \pm 0.46$ | $-2.31 \pm 0.67$ | $-0.31$ | 0.002 * |
| | 20°T | $-2.51 \pm 0.69$ | $-2.78 \pm 0.85$ | $-0.27$ | 0.009 * |
| 5.0 D | Axial | $-4.49 \pm 0.50$ | $-4.45 \pm 0.79$ | 0.04 | 0.973 |
| | 10°N | $-4.30 \pm 0.61$ | $-4.55 \pm 0.72$ | $-0.25$ | 0.017 * |
| | 20°N | $-4.20 \pm 0.77$ | $-4.51 \pm 0.94$ | $-0.31$ | 0.004 * |
| | 10°T | $-4.78 \pm 0.57$ | $-5.07 \pm 0.70$ | $-0.29$ | 0.013 * |
| | 20°T | $-5.30 \pm 0.82$ | $-5.64 \pm 0.89$ | $-0.34$ | 0.003 * |

[§] Wilcoxon test. * Statistically significant.

The Friedman test was used to analyze the differences in mean spherical equivalent with eccentricity, disclosing statistically significant differences (*p*-value < 0.001) for all examination conditions. Results obtained with the pair-wise Conover's post hoc test are presented in Table 2.

A myopic increase in nasal and temporal eccentricity was observed in M, with statistically significant differences at 10° and 20° in the temporal retina and 20° in the nasal retina. A statistically significant nasal-temporal asymmetry in M was found at 10° and 20° of eccentricity, presenting more negative values in the temporal retina in all accommodation and light conditions.

**Table 2.** Pair-wise differences in spherical equivalent according to eccentricity for each accommodative demand and illumination (R: red; W: white).

| Compared Conditions | | *p*-Values [§] | | | |
|---|---|---|---|---|---|
| | | 2.5 D_W | 5.0 D_W | 2.5 D_R | 5.0 D_R |
| 0° | 10°N | 0.355 | 0.179 | 0.933 | 0.967 |
| | 20°N | 0.034 * | 0.066 | 0.181 | 0.834 |
| | 10°T | <0.001 * | 0.013 * | 0.004 * | <0.001 * |
| | 20°T | <0.001 * | <0.001 * | <0.001 * | <0.001 * |
| 10°N | 20°N | 0.223 | 0.613 | 0.209 | 0.867 |
| | 10°T | <0.002 * | <0.001 * | 0.006 * | <0.001 * |
| | 20°T | <0.001 * | <0.001 * | <0.001 * | <0.001 * |
| 20°N | 10°T | 0.045 * | <0.001 * | 0.122 | <0.001 * |
| | 20°T | <0.001 * | <0.001 * | <0.001 * | <0.001 * |
| 10°T | 20°T | 0.014 * | 0.033 * | 0.028 * | 0.025 * |

[§] Conover test. * Statistically significant.

### 3.3. Astigmatic Vector Components (J0 and J45)

Figure 6 plots the mean of the astigmatic vector components J0 (or vertical/horizontal astigmatism) and J45 (or oblique astigmatism) for each eccentricity, accommodative demand and illumination condition.

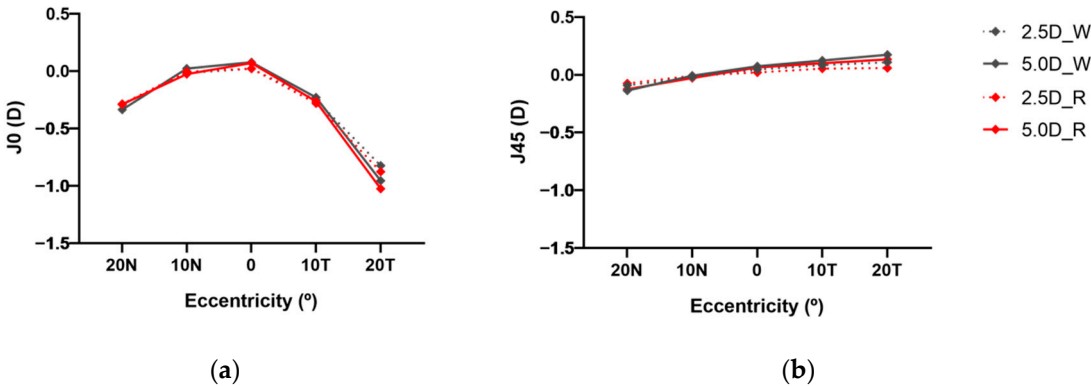

(**a**)                                                                         (**b**)

**Figure 6.** Astigmatic vector components at each eccentricity for 2.5 D and 5 D accommodative demands. (**a**) J0; (**b**) J45. (W: white light; R: red light; N: nasal retina; T: temporal retina).

The magnitude of J0 shows a nasal-temporal asymmetry, with a more marked decrease in the temporal retina, and with an increase in against the rule astigmatism. The oblique astigmatism component J45 also presents a nasal-temporal asymmetry, with more positive values in the temporal retina, although of smaller magnitude. Descriptive data are summarized in Table 3 and the results of the Wilcoxon test examining the differences of J0 and J45 between the two illumination conditions are presented. No statistically significant differences in the astigmatic vector components were found between white and red light in any examination condition.

Table 4 presents the analysis of the differences in J0 and J45 with accommodation. Some statistically significant differences were observed in the most peripheral gaze positions, although these differences were not clinically significant.

The Friedman test was used to analyze the differences in J0 and J45 with eccentricity. Eccentricity had a significant effect on J0 for all examination conditions (*p*-value < 0.001), as well as on J45 for all examination conditions except for 2.5 D of accommodation and red light illumination. Table 5 displays the results of Conover's post hoc test. J0 adopted negative values, with a statistically significant increase in eccentricity, as compared to the central value. A nasal-temporal significant asymmetry was observed, with more negative

values in the temporal retina. J45 was found to present negative values in the nasal retina and positive values towards the periphery of the temporal retina.

**Table 3.** Astigmatic vector components (mean $\pm$ standard deviation) for both illumination conditions (red and white light) at each eccentricity and accommodative demand. (N: nasal retina; T: temporal retina).

| Astigmatic Vector Components (Mean $\pm$ SD) | | | | | | | |
|---|---|---|---|---|---|---|---|
| | | J0 | | | J45 | | |
| Examination Conditions | | White Light | Red Light | *p*-Value [§] | White Light | Red Light | *p*-Value [§] |
| 2.5 D | Axial | $0.021 \pm 0.060$ | $0.022 \pm 0.104$ | 0.838 | $0.046 \pm 0.061$ | $0.020 \pm 0.070$ | 0.245 |
| | 10°N | $-0.016 \pm 0.116$ | $-0.004 \pm 0.137$ | 0.536 | $-0.020 \pm 0.106$ | $-0.008 \pm 0.096$ | 0.239 |
| | 20°N | $-0.293 \pm 0.221$ | $-0.285 \pm 0.222$ | 0.341 | $-0.091 \pm 0.178$ | $-0.074 \pm 0.188$ | 0.453 |
| | 10°T | $-0.251 \pm 0.177$ | $-0.281 \pm 0.176$ | 0.380 | $0.084 \pm 0.112$ | $0.054 \pm 0.164$ | 0.380 |
| | 20°T | $-0.824 \pm 0.219$ | $-0.877 \pm 0.264$ | 0.056 | $0.109 \pm 0.193$ | $0.059 \pm 0.291$ | 0.149 |
| 5.0 D | Axial | $0.078 \pm 0.101$ | $0.071 \pm 0.129$ | 0.857 | $0.075 \pm 0.080$ | $0.065 \pm 0.075$ | 0.477 |
| | 10°N | $0.023 \pm 0.137$ | $-0.026 \pm 0.165$ | 0.580 | $-0.008 \pm 0.120$ | $-0.028 \pm 0.151$ | 0.580 |
| | 20°N | $-0.336 \pm 0.217$ | $-0.289 \pm 0.258$ | 0.406 | $-0.136 \pm 0.196$ | $-0.125 \pm 0.223$ | 0.406 |
| | 10°T | $-0.229 \pm 0.127$ | $-0.264 \pm 0.165$ | 0.280 | $0.124 \pm 0.129$ | $0.101 \pm 0.213$ | 0.280 |
| | 20°T | $-1.026 \pm 0.237$ | $-0.956 \pm 0.283$ | 0.175 | $0.175 \pm 0.261$ | $0.135 \pm 0.317$ | 0.175 |

[§] Wilcoxon test.

**Table 4.** Differences of J0 and J45 with accommodation at each examination condition.

| | | *p*-Values [§] (2.5 D–5 D) | |
|---|---|---|---|
| Examination Conditions | | J0 | J45 |
| Red light | Axial | 0.061 | 0.016 * |
| | 10°N | 0.696 | 0.981 |
| | 20°N | 0.018 * | 0.060 |
| | 10°T | 0.839 | 0.053 |
| | 20°T | <0.001 * | 0.003 * |
| White light | Axial | 0.009 * | 0.121 |
| | 10°N | 0.118 | 0.876 |
| | 20°N | 0.010 * | 0.065 |
| | 10°T | 0.614 | 0.034 * |
| | 20°T | 0.007 * | 0.004 * |

[§] Wilcoxon test. * Statistically significant.

**Table 5.** Pair-wise differences in astigmatic components J0 and J45 according to eccentricity for each accommodative demand and illumination (R: red; W: white).

| *p*-Values [§] | | | | | | | | | |
|---|---|---|---|---|---|---|---|---|---|
| | | J0 | | | | J45 | | | |
| Compared Conditions | | 2.5 D_W | 5.0 D_W | 2.5 D_R | 5.0 D_R | 2.5 D_W | 5.0 D_W | 2.5 D_R | 5.0 D_R |
| 0° | 10°N | 0.502 | 0.529 | 0.704 | 0.123 | 0.101 | 0.222 | 0.736 | 0.256 |
| | 20°N | <0.001 * | <0.001 * | <0.001 * | <0.001 * | 0.008 * | 0.002 * | 0.112 | 0.008 * |
| | 10°T | <0.001 * | <0.001 * | <0.001 * | <0.001 * | 0.582 | 0.933 | 0.966 | 0.768 |
| | 20°T | <0.001 * | <0.001 * | <0.001 * | <0.001 * | 0.291 | 0.448 | 0.736 | 0.899 |
| 10°N | 20°N | <0.001 * | <0.001 * | <0.001 * | 0.002 * | 0.291 | 0.049 * | 0.208 | 0.121 |
| | 10°T | <0.001 * | <0.001 * | <0.001 * | <0.001 * | 0.029 * | 0.192 | 0.768 | 0.399 |
| | 20°T | <0.001 * | <0.001 * | <0.001 * | <0.001 * | 0.008 * | 0.049 * | 1.000 | 0.207 |
| 20°N | 10°T | 1.000 | 0.335 | 1.000 | 0.645 | 0.001 * | 0.001 * | 0.122 | 0.018 * |
| | 20°T | <0.001 * | 0.007 * | 0.001 * | <0.001 * | <0.001 * | <0.001 * | 0.208 | 0.005 * |
| 10°T | 20°T | <0.001 * | <0.001 * | 0.001 * | <0.001 * | 0.611 | 0.500 | 0.768 | 1.000 |

[§] Conover test. * Statistically significant.

### 3.4. On-Axis Accommodative Response

Mean relative spherical values with reference to the baseline on-axis outcome measured without accommodation were analyzed to determine the accommodative response in different conditions of accommodative demand and illumination (Table 6).

**Table 6.** Accommodative response (mean $\pm$ SD) in red and white light.

| | Accommodative Response (Mean $\pm$ SD) | | | |
|---|---|---|---|---|
| Accommodative Demand | White Light | Red Light | Difference $_{(W-R)}$ | *p*-Value $^{\S}$ |
| 2.5 D | $-1.64 \pm 0.52$ | $-1.91 \pm 0.54$ | $-0.27$ | 0.017 * |
| 5.0 D | $-4.43 \pm 0.50$ | $-4.41 \pm 0.74$ | 0.02 | 0.866 |

$^{\S}$ *t*-Student test. * Statistically significant.

Figure 7 displays the induced on-axis accommodation as a function of accommodative demand, also showing the theoretical values, where the expected induced accommodation is equal to the accommodative demand. A greater accommodative response was observed under red light illumination at 2.5 D of accommodative demand. However, this finding was not observed at 5 D of accommodative demand, although, as noted above, greater sphere values were found in all other gaze positions.

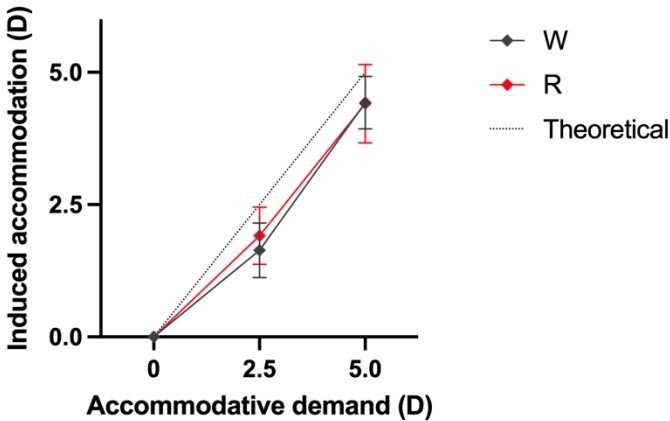

**Figure 7.** Mean on-axis induced accommodation for each accommodative demand and illumination (R: red light; W: white light; dashed line: theoretical expected values).

### 3.5. Relative Peripheral Refractive Error (RPRE)

Given the influence of illumination conditions on the accommodative response noted above, RPRE was only analyzed under white light. Table 7 summarizes RPRE for each eccentricity and accommodative demand. These findings are also shown in Figure 8.

**Table 7.** RPRE values for each eccentricity and accommodative demand (mean $\pm$ SD).

| | RPRE (Mean $\pm$ SD) | |
|---|---|---|
| Eccentricity | 2.5 D | 5.0 D |
| Axial | 0 | 0 |
| 10°N | $-0.041 \pm 0371$ | $0.195 \pm 0.397$ |
| 20°N | $-0.145 \pm 0.565$ | $0.297 \pm 0.536$ |
| 10°T | $-0.375 \pm 0.333$ | $-0.291 \pm 0.355$ |
| 20°T | $-0.886 \pm 0.489$ | $-0.813 \pm 0.621$ |

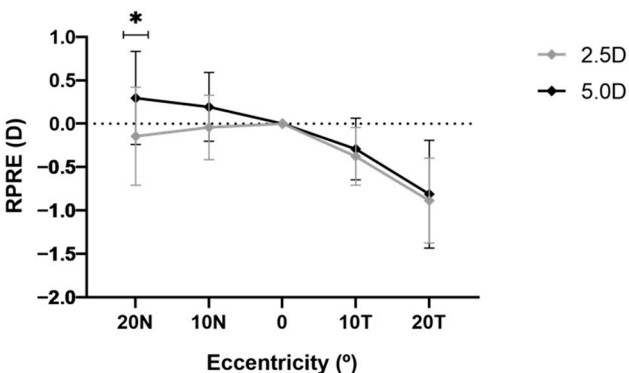

**Figure 8.** RPRE values for each eccentricity and accommodative demand. (* Statistically significant).

An ANOVA test revealed a statistically significant effect of both accommodation (*p*-value of 0.010) and eccentricity (*p*-value < 0.001) on RPRE. A pair-wise analysis with the Bonferroni post hoc test evidenced that RPRE assumes negative values with temporal eccentricity at 10° (*p*-value = 0.001) and at 20° (*p*-value < 0.001), with no statistically significant changes in accommodation. In contrast, a hyperopic shift was found in the nasal retina, with statistically significant differences between 2.5 D and 5 D at 20°.

## 4. Discussion

Most strategies aiming at myopia control are based on modifications of peripheral focus. Therefore, a full characterization of the response of the peripheral retina under various conditions of illumination and accommodation is imperative. The aim of this study was to explore low order aberrations (in terms of M, J0 and J45), as well as accommodative response and relative peripheral refractive error, at different eccentricities and under two illumination conditions (red and white light) and two accommodative demands (2.5 and 5 D).

The present findings evidenced an effect of illumination on various parameters. Thus, the mean difference in M between red and white illumination was $-0.28 \pm 0.02$ D and $-0.23 \pm 0.15$ D for accommodative demands of 2.5 D and 5 D, respectively. This effect accounts for the variation of the ocular focal length under a higher wavelength illumination, which leads to an increment in accommodative demand. This difference in M may be considered clinically significant, considering that refractive error is commonly measured in 0.25 D steps, and was also observed in a greater accommodative response under red light conditions at 2.5 D of accommodative demand. In contrast, illumination did not evidence any significant effect on J0 and J45. Therefore, whereas the mild changes in focus associated with accommodative response influenced the spherical component of the refraction, they did not have a significant effect on the astigmatic components [35,36].

Regarding eccentricity, a myopic, albeit asymmetric, increase in M was found in the temporal and nasal retina, with more negative values in the temporal retina in all experimental conditions. Similar nasal-temporal asymmetries were found in the J0 component, in agreement with previous research [37], and, on a smaller scale, in the J45 oblique astigmatism. The relative peripheral refractive error also displayed an asymmetrical behavior, with negative values at 10° and 20° in the temporal retina and positive values in the nasal retina, with a larger hyperopic shift for 5 D of accommodative demand. These asymmetries may reflect differences in anatomical structures of the ocular fundus and eye globe and are relevant when considering myopia control strategies, which rely on symmetrical and annular approaches to peripheral defocus, such as those created by orthokeratology and myopia control ophthalmic or soft contact lenses [14–18]. Such rotationally symmetric designs might, however, induce asymmetric refractive effects due to decentration. Furthermore, the posterior retinal contour of the eye is not necessarily symmetric in opposite quadrants, particularly in myopic eyes showing posterior ocular stretching [16,38]. In addition, it is interesting to note that research on changes in peripheral refraction with myopia

control strategies is commonly conducted in conditions without accommodation. As the aim of these strategies is to induce a myopic peripheral defocus, it may be assumed that accommodation would reinforce this effect. It would be relevant to design experimental conditions to test this hypothesis, that is, to measure peripheral refraction under myopia controlling strategies (orthokeratology, soft contact lenses and ophthalmic lenses) and with the presence of accommodative stimuli of different magnitude.

Another interesting result is the hyperopic shift observed in the nasal retina for the near accommodation target (Figure 8). Whatham et al. [27] and Queirós et al. [39] found also significant changes in the more eccentric retina for increasing accommodation demand. However, in those experiments, they found an opposite trend, with the spherical equivalent shifting in the myopic rather than hyperopic direction. There are methodological differences that can justify these opposite results. The instruments for measurement were different and while their studies analyzed more peripheral locations, the present study investigated the peripheral refraction up to 20° of eccentricity. However, this controversial result warrants further research to investigate whether parafoveal and peripheral focusing changes during intense near work can act as a stimulus for myopic progression.

This study was not devoid of limitations. Firstly, for the purposes of the study, the sample only included healthy young emmetropic participants and, therefore, results may not be extrapolated to populations with different characteristics, in particular to older participants with a reduced amplitude of accommodation. Secondly, for both accommodative distances, the fixation targets were placed at the same plane, in contrast to previous studies, which mounted the targets on a spherically curved surface to maintain the same accommodation for all the off-axis angles [26]. Thirdly, the experimental design was limited to the horizontal visual field and to a relatively narrow range of peripheral gaze positions, up to 20° of eccentricity. Some authors have suggested that significant changes with accommodation occur beyond 30° of eccentricity [27]. Other authors have also analyzed the ocular aberrations along the vertical meridian [20,25], providing a more complete representation of the image formation in the retina. Finally, all measurement sessions followed a predefined and constant order of presentation, which may not exclude a possible effect of visual fatigue on induced accommodation in the last set of measurements. For instance, Davies and Mallen randomized both the accommodative distance and the angle of presentation for each subject [28]. It must also be noted that a recent publication by Romashchenko and co-workers describes an experimental design allowing a more precise RPRE calculation through simultaneous measurement of foveal and peripheral aberrations with accommodation [29]. This experimental setup also permits binocular visualization of the targets, thus creating natural accommodation conditions.

Future research may consider implementing additional biometric measurements in order to evaluate the actual symmetry of the eye globe within the framework of the encountered asymmetries in the refractive components analyzed in this study. In addition, a full aberrometric characterization of the peripheral retina, including higher order aberrations, is essential to better understand the visual contribution of this part of the retina. The results of this ongoing research shall be the subject of a future publication.

## 5. Conclusions

The M component was myopic at both accommodative demands and showed a statistically significant myopic increase under red light conditions. There was little change on J0 and J45 with accommodation and no change between the illumination conditions. However, J0 increased significantly with eccentricity, exhibiting a nasal-temporal asymmetry. The RPRE was myopic at both accommodation demands and showed a statistically significant hypermetropic shift at 20° in the nasal retina. More studies are necessary to develop statistical ocular models with the assistance of optical aberrometric patterns, including both low and high order aberrations, at different regions of the visual field and at different accommodative states.

**Author Contributions:** Conceptualization, R.v.G., J.M.G.-M. and G.C.; methodology, R.v.G. and M.M.; formal analysis, R.v.G. and G.C.; investigation, R.v.G. and M.M.; resources, J.M.G.-M., R.v.G. and M.M.; data curation, R.v.G. and M.M.; writing—original draft preparation, R.v.G., M.M. and G.C.; writing—review and editing, G.C. and J.M.G.-M.; visualization, R.v.G.; supervision, J.M.G.-M. and G.C.; project administration, R.v.G. and M.M. All authors have read and agreed to the published version of the manuscript.

**Funding:** This research was funded by the Agencia Estatal de Investigación, Ministerio de Ciencia e Innovación of the Spanish government (PID2020-114582RB-I00/AEI/10.13039/501100011033). This project has received funding from the European Union's Horizon 2020 research and innovation programme under the Marie Skłodowska-Curie grant agreement No 956720.

**Institutional Review Board Statement:** The study was conducted in accordance with the Declaration of Helsinki, and approved by an Institutional Review Board.

**Informed Consent Statement:** Informed consent was obtained from all subjects involved in the study.

**Data Availability Statement:** The datasets generated and analyzed during the study are available from the corresponding authors upon reasonable request.

**Conflicts of Interest:** The authors declare no conflict of interest.

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
