# Peer review of "The Effect of Accommodation on Peripheral Refraction under Two Illumination Conditions"

_photonics, doi:10.3390/photonics9050364_

Round 1

Reviewer 1 Report

In this paper, the changes in relative peripheral refraction in emmetropic eyes are measured during reading. The measurements were made under two light conditions and two different accommodation requirements. The results showed an asymmetric RPR with more relative myopia in the temporal retina compared to the nasal retina. At 5 dpt accommodation effort, there was also relative hyperopic refraction in the nasal retina.  

The study takes up a much debated approach in the myopia research community. Does refraction in the peripheral retina have an impact on myopia progression - yes or no? 
I find measuring the RPR under accommodation condition quite interesting. Therefore, I read this paper with great interest. 

Thank you very much, from my point of view a very successful and clearly structured paper. 

What I would be additionally interested in and what I find missing in the discussion is your opinion on how you relate your results to the currently discussed anti-myopia lenses.  Your results (for emmetropia and not for myopia participants) show a relative myopia in the periphery during accommodation. The lens manufacturers promise to slow down myopia by producing a relative myopia. Based on your results, accommodation is already an anti-myopia therapy? If you could briefly include this question in the discussion and comment on it. 

Author Response

Reviewer #1

In this paper, the changes in relative peripheral refraction in emmetropic eyes are measured during reading. The measurements were made under two light conditions and two different accommodation requirements. The results showed an asymmetric RPR with more relative myopia in the temporal retina compared to the nasal retina. At 5 dpt accommodation effort, there was also relative hyperopic refraction in the nasal retina.  

The study takes up a much-debated approach in the myopia research community. Does refraction in the peripheral retina have an impact on myopia progression - yes or no? 
I find measuring the RPR under accommodation condition quite interesting. Therefore, I read this paper with great interest. 

Thank you very much, from my point of view a very successful and clearly structured paper. 

RESPONSE: We would like to express our gratitude to the reviewer for his / her kind words of praise.

What I would be additionally interested in and what I find missing in the discussion is your opinion on how you relate your results to the currently discussed anti-myopia lenses.  Your results (for emmetropia and not for myopia participants) show a relative myopia in the periphery during accommodation. The lens manufacturers promise to slow down myopia by producing a relative myopia. Based on your results, accommodation is already an anti-myopia therapy? If you could briefly include this question in the discussion and comment on it. 

RESPONSE: Thank you for this comment. We have briefly discussed this interesting issue in the revised Discussion section of the manuscript (lines 335-339).

Reviewer 2 Report

The paper entitled “The effect of accommodation on peripheral refraction under two illumination conditions” is a study based on the development of an experimental design to measure on-axis and off-axis low-order optical aberrations at different levels of accommodation. The manuscript is interesting, innovative, and of potential clinical interest.

The results show that The M component was myopic at both accommodative demands and showed a statistically significant myopic increase under red light conditions. Moreover, the RPRE was myopic at both accommodation demands and showed a statistically significant hypermetropic shift at 20° in the nasal retina.

The study has been correctly planned. It is well written and of clinical interest. The study provides objective results, which adds to current literature in this field.

There are, however, several issues that need to be addressed by the authors, which include:

  1. The age range should be included in the selection criteria. It would have been interesting to see the results of older subjects, which could be included as a minor limit to the study.
  2. Further mention should be made as to how the results can be applied in a clinical setting, and how specific future studies could address factors that still need further investigation.
  3. The English can be improved for better flow.

Author Response

Reviewer #2

The paper entitled “The effect of accommodation on peripheral refraction under two illumination conditions” is a study based on the development of an experimental design to measure on-axis and off-axis low-order optical aberrations at different levels of accommodation. The manuscript is interesting, innovative, and of potential clinical interest.

The results show that The M component was myopic at both accommodative demands and showed a statistically significant myopic increase under red light conditions. Moreover, the RPRE was myopic at both accommodation demands and showed a statistically significant hypermetropic shift at 20° in the nasal retina.

The study has been correctly planned. It is well written and of clinical interest. The study provides objective results, which adds to current literature in this field.

RESPONSE: Thank you. We are happy to learn that the manuscript was found to be interesting and the study well-designed and resulting in clinically relevant information.

There are, however, several issues that need to be addressed by the authors, which include:

1. The age range should be included in the selection criteria. It would have been interesting to see the results of older subjects, which could be included as a minor limit to the study.

RESPONSE: Indeed. Although the aim of this study was to explore peripheral refraction under accommodation in young and healthy subjects, further research with older participants would provide additional information on the actual role of the accommodative response. We have added information to the inclusion criteria of our revised manuscript (line 72) and briefly addressed this limitation in the Discussion section (line 355-356).

2. Further mention should be made as to how the results can be applied in a clinical setting, and how specific future studies could address factors that still need further investigation.

RESPONSE: We have added an analysis of the possible implications of peripheral myopic changes in accommodation (lines 335-339) and suggested additional avenues for future research (lines 339-342).

3. The English can be improved for better flow.

RESPONSE: We have revised the whole manuscript and edited some sentences for clarity.

Reviewer 3 Report

The authors carefully performed an experiment to measure on-axis and off-axis low order optical aberrations with different settings in healthy subjects. It is a well-organized study withwith comprehensive ocular examiniation. The study presented novel and important data and insightful interpretation. The only concern for is that, how did the findings benifit to the patients? I would suggest the authors to interprete more on this. 

Author Response

Reviewer #3

The authors carefully performed an experiment to measure on-axis and off-axis low order optical aberrations with different settings in healthy subjects. It is a well-organized study with comprehensive ocular examination. The study presented novel and important data and insightful interpretation.

RESPONSE: Thank you for these comments.

The only concern for is that, how did the findings benefit to the patients? I would suggest the authors to interpret more on this. 

RESPONSE: Following the suggestion of all reviewers, we have further explored the clinical implications of the present findings (lines 335-342). Thank you.